# Genetic dissection of mutual interference between two consecutive learning tasks in *Drosophila*

Jianjian Zhao[1,2], Xuchen Zhang[1,2†], Bohan Zhao[1,2‡], Wantong Hu[1,2], Tongxin Diao[1,2], Liyuan Wang[1,2], Yi Zhong[1,2], Qian Li[1,2*]

[1]School of Life Sciences, IDG/McGovern Institute for Brain Research, MOE Key Laboratory of Protein Sciences, Tsinghua University, Beijing, China; [2]Tsinghua-Peking Center for Life Sciences, Beijing, China

**\*For correspondence:**
liqian8@tsinghua.edu.cn

**Present address:** †Department of Molecular and Cellular Physiology and Howard Hughes Medical Institute, Stanford University School of Medicine, Stanford, United States; ‡Department of Neuroscience, Dorris Neuroscience Center, Scripps Research, San Diego, United States

**Competing interest:** The authors declare that no competing interests exist.

**Abstract** Animals can continuously learn different tasks to adapt to changing environments and, therefore, have strategies to effectively cope with inter-task interference, including both proactive interference (Pro-I) and retroactive interference (Retro-I). Many biological mechanisms are known to contribute to learning, memory, and forgetting for a single task, however, mechanisms involved only when learning sequential different tasks are relatively poorly understood. Here, we dissect the respective molecular mechanisms of Pro-I and Retro-I between two consecutive associative learning tasks in *Drosophila*. Pro-I is more sensitive to an inter-task interval (ITI) than Retro-I. They occur together at short ITI (<20 min), while only Retro-I remains significant at ITI beyond 20 min. Acutely overexpressing Corkscrew (CSW), an evolutionarily conserved protein tyrosine phosphatase SHP2, in mushroom body (MB) neurons reduces Pro-I, whereas acute knockdown of CSW exacerbates Pro-I. Such function of CSW is further found to rely on the γ subset of MB neurons and the downstream Raf/MAPK pathway. In contrast, manipulating CSW does not affect Retro-I as well as a single learning task. Interestingly, manipulation of Rac1, a molecule that regulates Retro-I, does not affect Pro-I. Thus, our findings suggest that learning different tasks consecutively triggers distinct molecular mechanisms to tune proactive and retroactive interference.

## Editor's evaluation

This fundamental study substantially advances our understanding of interactions of consecutive memory tasks by identifying responsible molecules and neurons. The evidence supporting the claims of the authors is solid. The work will be of broad interest to neuroscientists working on learning and memory as well as learning psychologists.

## Introduction

Continual learning is a natural ability of animals ranging from invertebrates to vertebrates but a great challenge for artificial intelligence (*Fayek et al., 2020*; *Kudithipudi et al., 2022*; *Parisi et al., 2019*; *Wang et al., 2023*). To achieve continual learning, the mutual interference between learning tasks, including proactive and retroactive interference, needs to be properly tuned. The interference from the previous task on the learning and memory of the current task is called Pro-I, while the interference from the current task on the memory of the following task is named Retro-I (*Bouton, 1993*; *Miller, 2021*; *Wixted, 2004*). Although, we have much understanding of the molecular mechanisms of learning and memory for a single learning task (*Johansen et al., 2011*; *Kandel et al., 2014*), the molecular mechanisms that modulate the interferences between different tasks remain unclear.

*Drosophila* is a well-studied and highly tractable genetic model organism for understanding molecular mechanisms underlying a single learning task (*Davis, 2005*; *Noyes et al., 2021*; *Waddell and Quinn, 2001*) as well as related human diseases (*Mariano et al., 2020*; *Pandey and Nichols, 2011*; *van Alphen and van Swinderen, 2013*). In recent years, *Drosophila* has also emerged as an excellent model for studying interference mechanisms between two associative learning tasks. Several molecules, including Rac1, Foraging, Scribble, SLC22A, Fmr1, SCAR, and Dia, have been reported to regulate Retro-I (*Cervantes-Sandoval et al., 2016*; *Dong et al., 2016*; *Gai et al., 2016*; *Gao et al., 2019*; *Reaume et al., 2011*; *Shuai et al., 2010*). Of these molecules, Scribble and SLC22A, have also been reported to regulate Pro-I (*Cervantes-Sandoval et al., 2016*; *Gai et al., 2016*). It indicates that Retro-I and Pro-I may have shared regulatory molecules. Previous studies reported that resistance to Pro-I and Retro-I is very different in patients with autism (*Mottron et al., 1998*), schizophrenia (*Torres et al., 2001*), and ADHD (*Orban et al., 2022*). It raises a possibility that Pro-I and Retro-I may have distinct molecular mechanisms. Of note, all reported molecules in regulating Retro-I or Pro-I also play important roles in the learning or memory of a single learning task (*Cervantes-Sandoval et al., 2016*; *Dong et al., 2016*; *Gai et al., 2016*; *Gao et al., 2019*; *Reaume et al., 2011*; *Shuai et al., 2010*). Whether there are molecules specifically responsible for modulating memory interference without affecting a single learning task remains to be determined.

## Results

Psychological studies have shown that Pro-I is closely related to content similarity, context similarity, and the time interval between the proactive task and the target task (*Kliegl and Bäuml, 2021*). In *Drosophila*, it has been known that an aversive associative learning task produces significant Pro-I on another similar task immediately following (*Cervantes-Sandoval et al., 2016*; *Gai et al., 2016*). However, the requirements for the generation of such Pro-I have not been systematically investigated, such as content similarity, context similarity, and the time interval between tasks.

### Pro-I is affected by content similarity and context similarity between tasks

Consistent with previous studies (*Cervantes-Sandoval et al., 2016*; *Gai et al., 2016*), significant Pro-I was observed between two consecutive aversive tasks (*Figure 1A*; Associative group). This Pro-I phenomenon was still evident 1 hr later (*Figure 1—figure supplement 1A*) and was not affected by changing the order of the two tasks (*Figure 1—figure supplement 1B*). To reduce content similarity between tasks, we changed the proactive task into non-associative stimuli (*Mo et al., 2022*; *Tully and Quinn, 1985*; *Zhang et al., 2018*) or an appetitive learning task (*Liu et al., 2012*; *Perisse et al., 2013*; *Pribbenow et al., 2022*), which was less similar to the target task. With such changes, no significant Pro-I was observed (*Figure 1A and B*). Then, we changed the learning context of the two aversive tasks to reduce context similarity (*Figure 1C*). When both tasks were learned in the same context, the Pro-I was evident. In contrast, when the two tasks were learned in different contexts, the Pro-I was no longer observed. In addition, recent studies have found that a single aversive learning task can produce two opposing memory components: avoidance memory and approach memory (*Jacob and Waddell, 2020*; *Naganos et al., 2022*; *Zhao et al., 2021*). Here, we found that the Pro-I only affected avoidance memory (*Figure 1—figure supplement 1C*). Together, these data suggest that reducing inter-task similarity by changing learning content or context can release Pro-I in *Drosophila*, which is consistent with psychological studies (*Kliegl and Bäuml, 2021*).

### Pro-I and Retro-I differ in time interval sensitivity and regulatory molecules

We next tested the relationship between the Pro-I and ITI (*Figure 2A–C*). Flies showed significant Pro-I when the ITI was 15 min or less (0, 5, 10, and 15 min). If the ITI was 20 min or more (20, 30, and 60 min), no significant Pro-I was observed. In contrast, when the ITI was gradually increased from 0 to 60 min (0, 20, 30, and 60 min), the flies consistently exhibited significant Retro-I (*Figure 2D and E*). This result is consistent with previous studies on Retro-I (*Cervantes-Sandoval et al., 2016*; *Dong et al., 2016*; *Gai et al., 2016*; *Gao et al., 2019*; *Reaume et al., 2011*; *Shuai et al., 2010*).

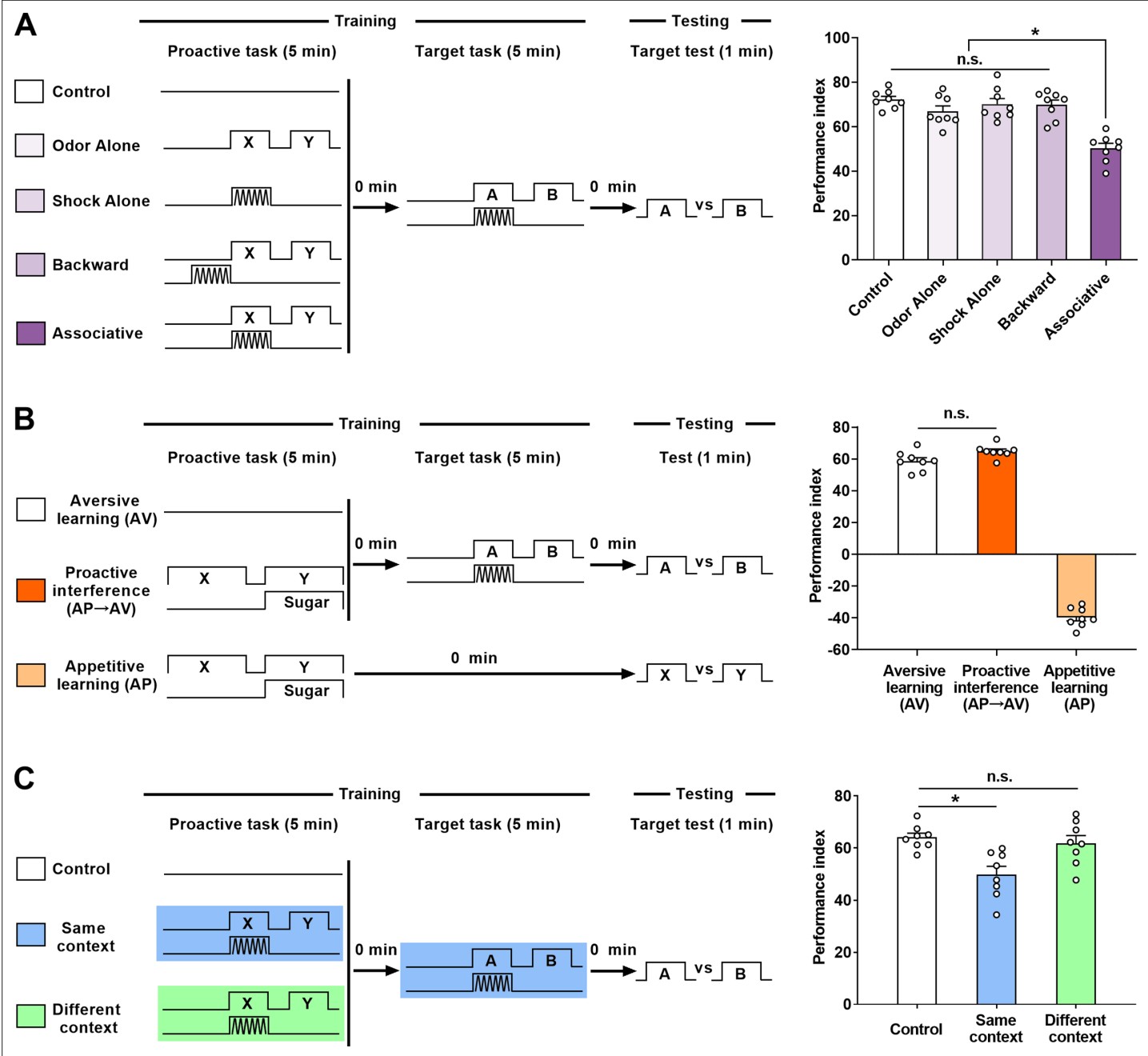

**Figure 1.** Effects of proactive interference (Pro-I) between two consecutive olfactory learning tasks. (**A**) Left: schematic of the experiment. No stimuli (Control), non-associative stimuli (Odor Alone, Shock Alone, and Backward), or associative learning (Associative) were used as the proactive task. Aversive associative learning was used as the target task. Right: Comparison of immediate memory performance of the target task of different groups. Compared with the 'Control' group, significant Pro-I was observed in the 'Associative' group, but not in non-associative groups. n=8. (**B**) Left: schematic of the experiment. In the 'Aversive learning (AV)' or 'Appetitive learning (AP)' group, the immediate performance of a single task was tested; in the 'Proactive interference (AP→AV)' group, the proactive task was an appetitive learning and the target task was an aversive learning. Right: Comparison of immediate memory performance of different groups. Compared with the AV group, no significant Pro-I was observed in the 'Proactive interference (AP→AV)' group. n=8. (**C**) Left: schematic of the experiment. In the 'Control' group, there was no proactive task; in the 'Same context' group, the proactive task and the target task were performed in the same context (blue light); in the 'Different context' group, the proactive task was performed in the green light context, while the target task was performed in the blue light context. Right: Comparison of immediate memory performance of the target task of different groups. Compared with the 'Control' group, significant Pro-I was observed in the 'Same context' group, but not in the 'Different context' group. n=8. Statistics: ordinary one-way ANOVA with Dunnett's multiple comparisons tests. Results with error bars are means ± SEM. *$p<0.05$. n.s., non-significant. Also see *Figure 1—figure supplement 1*, *Figure 1—source data 1*, and *Figure 1—figure supplement 1—source data 1* information.

*Figure 1 continued on next page*

*Figure 1 continued*

The online version of this article includes the following source data and figure supplement(s) for figure 1:

**Source data 1.** Raw data of *Figure 1*.

**Figure supplement 1.** Other effects of proactive interference (Pro-I).

**Figure supplement 1—source data 1.** Raw data of *Figure 1—figure supplement 1*.

Given that Scribble and SLC22A regulate both Pro-I and Retro-I (*Cervantes-Sandoval et al., 2016*; *Gai et al., 2016*), other regulators of Retro-I may also affect Pro-I. Rac1 is required to mediate Retro-I when the ITI is 1.5 hr (*Shuai et al., 2010*). We next tested whether Rac1 also affects Retro-I with 0 min ITI and explored whether Rac1 plays a role in Pro-I (*Figure 2F*). The transgene dominant-negative Rac1 (Rac1-DN) or constitutively active Rac1 (Rac1-CA) was used to inhibit or increase Rac1 activity, respectively (*Luo et al., 1994*). Since acute manipulation of Rac1 activity in mushroom body (MB) neurons is sufficient to regulate memory performance (*Gao et al., 2019*; *Shuai et al., 2010*), Rac1 transgenes were expressed using MB-GS, a Gene-Switch (GS) tool capable of inducing transgene expression specifically in the MB only on administration of the drug RU486 (*Mao et al., 2004*). Suppressing (*MB-GS/UAS-Rac1-DN*, RU486+) or increasing (*MB-GS/UAS-Rac1-CA*, RU486+) Rac1 activity in MB neurons significantly mitigated or aggravated the Retro-I, respectively. However, the same manipulations did not affect the Pro-I, indicating that there is a different molecular mechanism underlying the Pro-I.

## Pro-I, but not Retro-I, is bidirectionally regulated by CSW

Since the difference between Pro-I and Retro-I lies in the sensitivity to inter-task time intervals, we speculate that molecules that modulate 'time interval' effects may specifically regulate Pro-I. Given that CSW, an evolutionarily conserved protein tyrosine phosphatase SHP2, has been reported

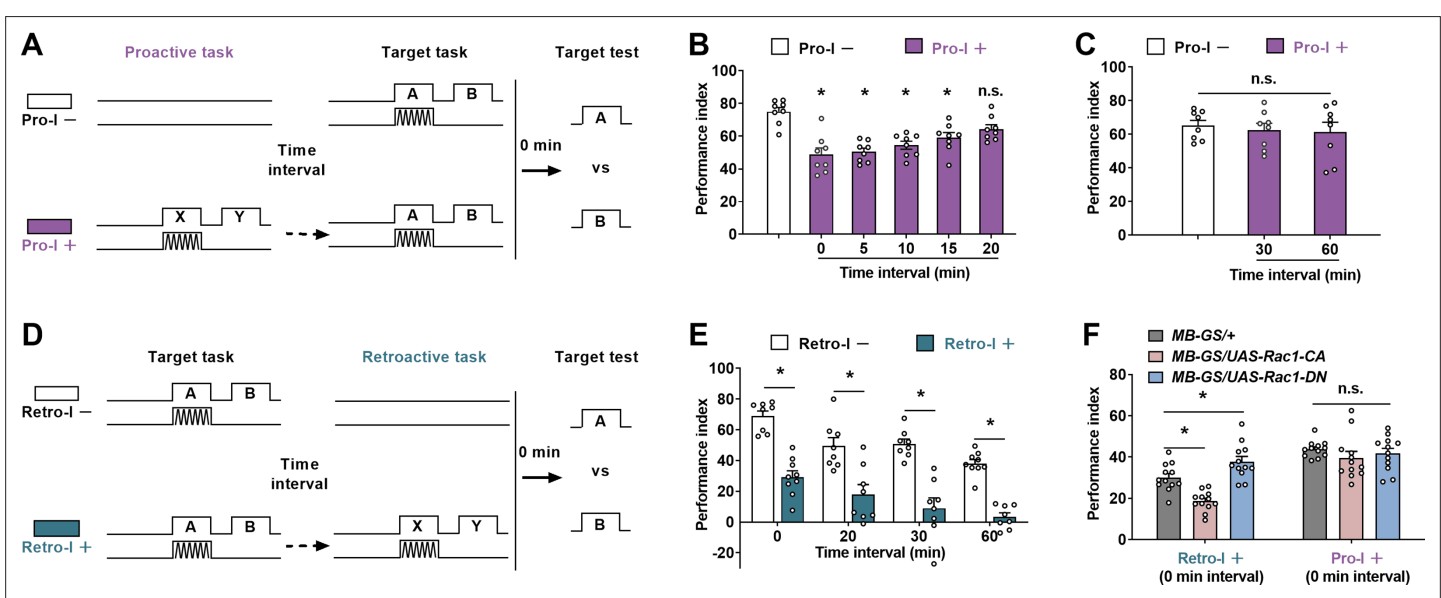

**Figure 2.** Differences between proactive interference (Pro-I) and retroactive interference (Retro-I). (**A–C**) The paradigm (**A**) and behavioral results (**B** and **C**) of Pro-I experiments. The time interval between the proactive task and the target task was changed from 0 to 60 min. Pro-I was significant when the time interval was less than 20 min (0 min, 5 min, 10 min, or 15 min) in wild-type flies. n=8. (**D** and **E**) The paradigm (**D**) and the behavioral result (**E**) of theRetro-I experiment. Retro-I was significant when the time interval between the target and the retroactive task was 0 min, 20 min, 30 min, and 60 min. n=8–9. (**F**) The behavioral performance of transgenic flies with retroactive or Pro-I. Compared to the control group (*MB-GS/+*, RU486+), Rac1-CA-expressing flies (*MB-GS/UAS-Rac1-CA*, RU486+) showed a significantly lower performance index, while Rac1-DN-expressing flies (*MB-GS/UAS-Rac1-DN*, RU486+) exhibited a higher memory index in Retro-I. No significant difference was observed in all groups with Pro-I. n=12. Statistics: ordinary one-way ANOVA with Dunnett's multiple comparisons tests (**B** and **C**); two-way ANOVA with Bonferroni's multiple comparisons tests (**E** and **F**). Results with error bars are means ± SEM. *p<0.05. n.s., non-significant. Also see *Figure 2—source data 1* for additional information.

The online version of this article includes the following source data for figure 2:

**Source data 1.** Raw data of *Figure 2*.

to regulate the 'time interval' effect in long-term memory (LTM) formation in *Drosophila* (*Pagani et al., 2009*), we tested the role of CSW in Pro-I. Compared to the parental control group (*MB-GS/+*, RU486+), acutely knocking down CSW in MB neurons using two independent RNAi lines (*MB-GS/ UAS-csw-RNAi-1* and *MB-GS/UAS-csw-RNAi-2*; RU486+) showed more severe Pro-I, which can be better reflected by the Pro-I+/Pro-I– ratio (*Figure 3A*). To further determine this result, we added a comparison with uninduced control groups (*Figure 3B*) or genetic control groups (*Figure 3—figure supplement 1A and 1D*) and obtained consistent results. When the ITI was 20 min, Pro-I was no longer observed in control flies but was still present in flies with acute knockdown of CSW (*Figure 3C*; *Figure 3—figure supplement 1B*). Conversely, acute overexpression of CSW in MB neurons (*MB-GS/ UAS-csw*, RU486+) reduced Pro-I by using a newly constructed transgenic strain (*Figure 3D and E*; *Figure 3—figure supplement 1C and E*) and a previously reported strain (*Botham et al., 2008*; *Figure 3—figure supplement 1F*). Of note, bidirectional manipulation of CSW expression in MB neurons did not affect Retro-I (*Figure 3F*).

The MB contains ~2000 intrinsic neurons called Kenyon cells (KCs), which are further divided into three major types: γ neurons (~675), α/β neurons (~990), and α'/β' neurons (~350) (*Aso et al., 2014*). We next sought to determine whether the modulatory effects of CSW on Pro-I could be narrowed to a certain type of MB neurons. 5-HT1B-Gal4 (*Gao et al., 2019*; *Shyu et al., 2017*; *Yuan et al., 2005*), VT30604-Gal4 (*Wu et al., 2013*), and C739-Gal4 (*O'Dell et al., 1995*) were used to drive transgene expression in γ neurons, α'/β' neurons, and α/β neurons, respectively. Overexpressing CSW in γ neurons or α/β neurons, but not in α'/β' neurons showed higher performance with Pro-I when compared with their respective genetic controls (*Figure 3—figure supplement 1G*). To rule out possible developmental effects, as there is no available GS tool for subtypes of MB neurons, we employed another inducible expression system called TARGET, which relies on a temperature shift to induce transgene expression (*McGuire et al., 2003*). Acute overexpression of CSW in γ neurons (*Gal80$^{ts}$/+; 5-HT1B/UAS-csw*, induced), but not in the α/β neurons (*C739/+; Gal80$^{ts}$/UAS-csw*, induced), significantly reduced the Pro-I compared with their respective controls (*Figure 3G and H*). In contrast, acute knockdown of CSW in γ neurons increased the Pro-I (*Figure 3I*). In uninduced flies, Pro-I was not affected (*Figure 3—figure supplement 1H*). These data suggest that CSW regulates Pro-I in MB γ neurons. In addition, the result in *Figure 1C* shows that no Pro-I occurred when the two tasks were learned in different contexts. In this case, knocking down the CSW remained ineffective (*Figure 3—figure supplement 1I*).

## The Raf/MAPK pathway acts downstream of the CSW to regulate Pro-I

CSW is generally considered as a positive regulator of Ras/MAPK signaling (*Perkins et al., 1996*). And the regulation of CSW on the 'time interval' effect in LTM formation is also thought to be via the MAPK pathway (*Pagani et al., 2009*). So, we further explored whether CSW affects Pro-I through the MAPK pathway. Feeding U0126, a widely used inhibitor of the MAPK pathway (*Thomas and Huganir, 2004*), did not affect single-task learning, but significantly exacerbated Pro-I when learning two consecutive tasks (*Figure 4A*). Flies with acute genetic knockdown of MAPK in MB neurons (*MB-GS/UAS-MAPK-RNAi*, RU486+) also exhibited more severe Pro-I than uninduced and parental control flies (*Figure 4B*; *Figure 4—figure supplement 1A*).

Consistent with these results, Raf kinase, a classical upstream regulator of MAPK (*Thomas and Huganir, 2004*), was found to bidirectionally regulate the Pro-I in MB neurons like CSW (*Figure 4C and D*). Acutely knocking down Raf significantly exacerbated the Pro-I, while acute overexpression of Raf-GOF, which encodes a constitutively active Raf kinase (*Brand and Perrimon, 1994*), significantly reduced Pro-I. When Raf-GOF and CSW-RNAi were co-expressed, Raf-GOF expression dominated the effect on Pro-I, indicating that Raf acts downstream of CSW (*Figure 4E*). Thus, our data support that CSW regulates Pro-I through Raf/MAPK pathway.

Our previous study found that the Raf/MAPK pathway is activated by learning to protect memory retention via non-muscle myosin Ⅱ Sqh after single-task learning (*Zhang et al., 2018*). Although the regulatory effect of CSW on Pro-I in two consecutive task learning was also through the Raf/MAPK pathway, manipulating CSW did not affect the learning and memory of a single task (*Figure 4—figure supplement 1B and C*), unlike Raf (*Figure 4—figure supplement 1D*; *Zhang et al., 2018*). Interestingly, as a downstream molecule of the Raf/MAPK pathway in regulating single-task memory (*Zhang et al., 2018*), Sqh did not participate in the Pro-I between the two tasks (*Figure 4—figure supplement*

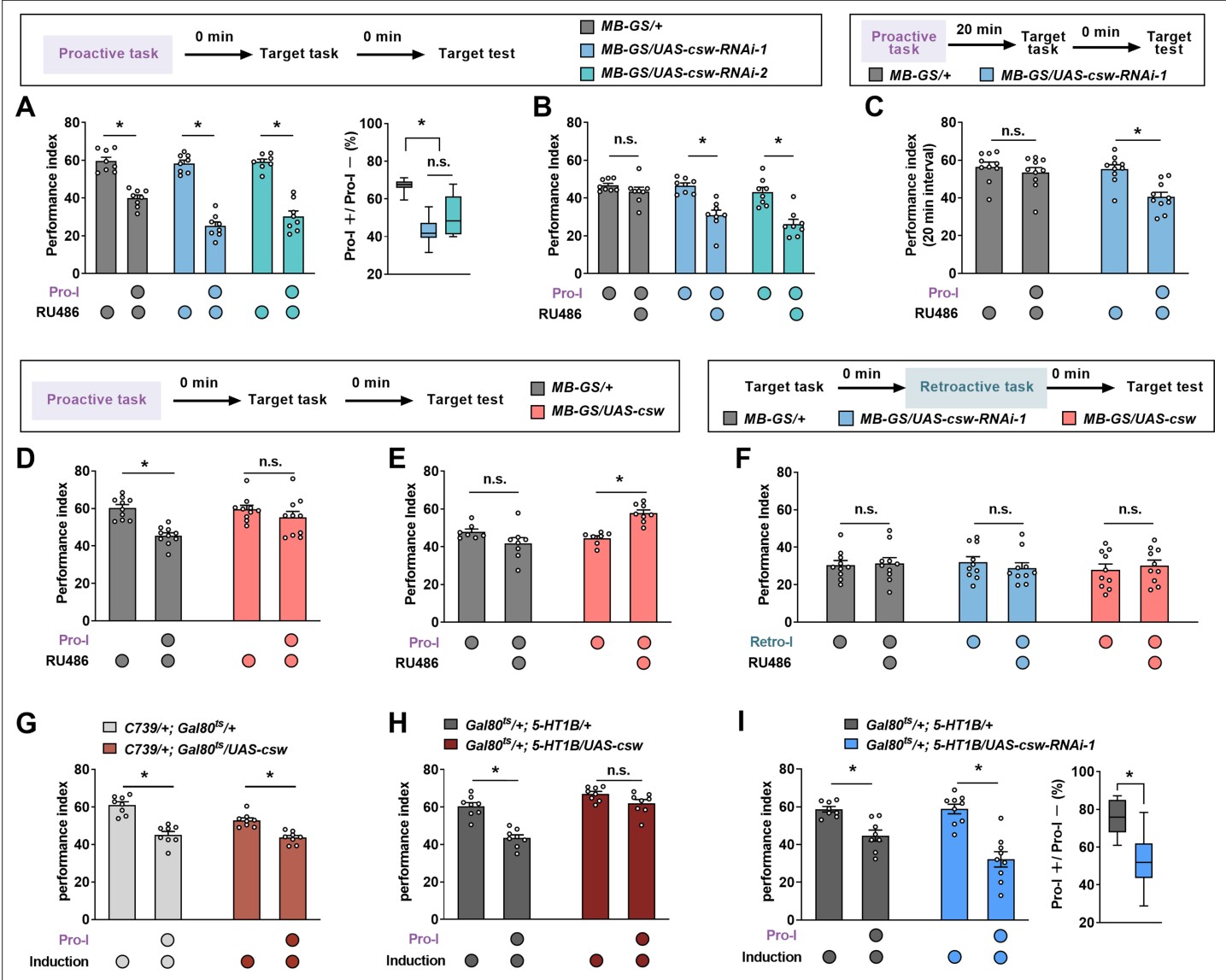

**Figure 3.** Corkscrew (CSW) bidirectionally regulates proactive but not retroactive interference (Retro-I). (**A and B**) The behavioral performance of transgenic flies with proactive interference (Pro-I) (0 min interval). Acute knockdown of CSW in mushroom body neurons (*MB-GS/UAS-csw-RNAi-1* or *MB-GS/UAS-csw-RNAi-2*; RU486+) led to more severe Pro-I relative to the genetic control group (*MB-GS/+*, RU486+) (**A**) and uninduced controls (RU486–) (**B**). n=8. (**C**) The behavioral performance of transgenic flies with Pro-I (20 min interval). csw-RNAi-expressing flies (*MB-GS/UAS-csw-RNAi-1*, RU486+) but not control flies (*MB-GS/+*, RU486+) showed significant Pro-I. n=10. (**D and E**) The behavioral performance of transgenic flies with Pro-I (0 min interval). Acute overexpression of CSW in mushroom body neurons (*MB-GS/UAS-csw*; RU486+) prevented Pro-I (**D**) and was more resistant to Pro-I than uninduced control (RU486–) (**E**). n=7–8. (**F**) The behavioral performance of transgenic flies with Retro-I (0 min interval). Acute knockdown (*MB-GS/UAS-csw-RNAi-1*, RU486+) or overexpression (*MB-GS/UAS-csw*, RU486+) of CSW in mushroom body neurons did not affect Retro-I compared with uninduced controls. n=10. (**G**) Significant Pro-I was found in flies with acute overexpression of CSW in MB α/β neurons (*C739/+; Gal80ts/UAS-csw*) and genetic control flies (*C739/+; Gal80ts/+*). n=8. (**H**) Significant Pro-I was found in genetic control flies (*Gal80ts/+; 5-HT1B/+*) but not flies with acute overexpression of CSW in MB γ neurons (*Gal80ts/+; 5-HT1B/UAS-csw*). n=8. (**I**) Acute knockdown of CSW in MB γ neurons (*Gal80ts/+; 5-HT1B/UAS-csw-RNAi-1*) increased Pro-I relative to the genetic control group (*Gal80ts/+; 5-HT1B/+*). n=8–9. Statistics: two-way ANOVA with Bonferroni's multiple comparisons tests (A-left panel, B, D-H, and I-left panel); Kruskal-Wallis test with Dunn's multiple comparisons tests (A-right panel); Mann-Whitney test (**C**); unpaired t-test (I-right panel). Results with error bars are means ± SEM. *p<0.05. n.s., non-significant. Also see *Figure 3—figure supplement 1*, *Figure 3—source data 1*, and *Figure 3—figure supplement 1—source data 1* for additional information.

The online version of this article includes the following source data and figure supplement(s) for figure 3:

**Source data 1.** Raw data of *Figure 3*.

**Figure supplement 1.** Additional control experiments of *Figure 3*.

**Figure supplement 1—source data 1.** Raw data of *Figure 3—figure supplement 1*.

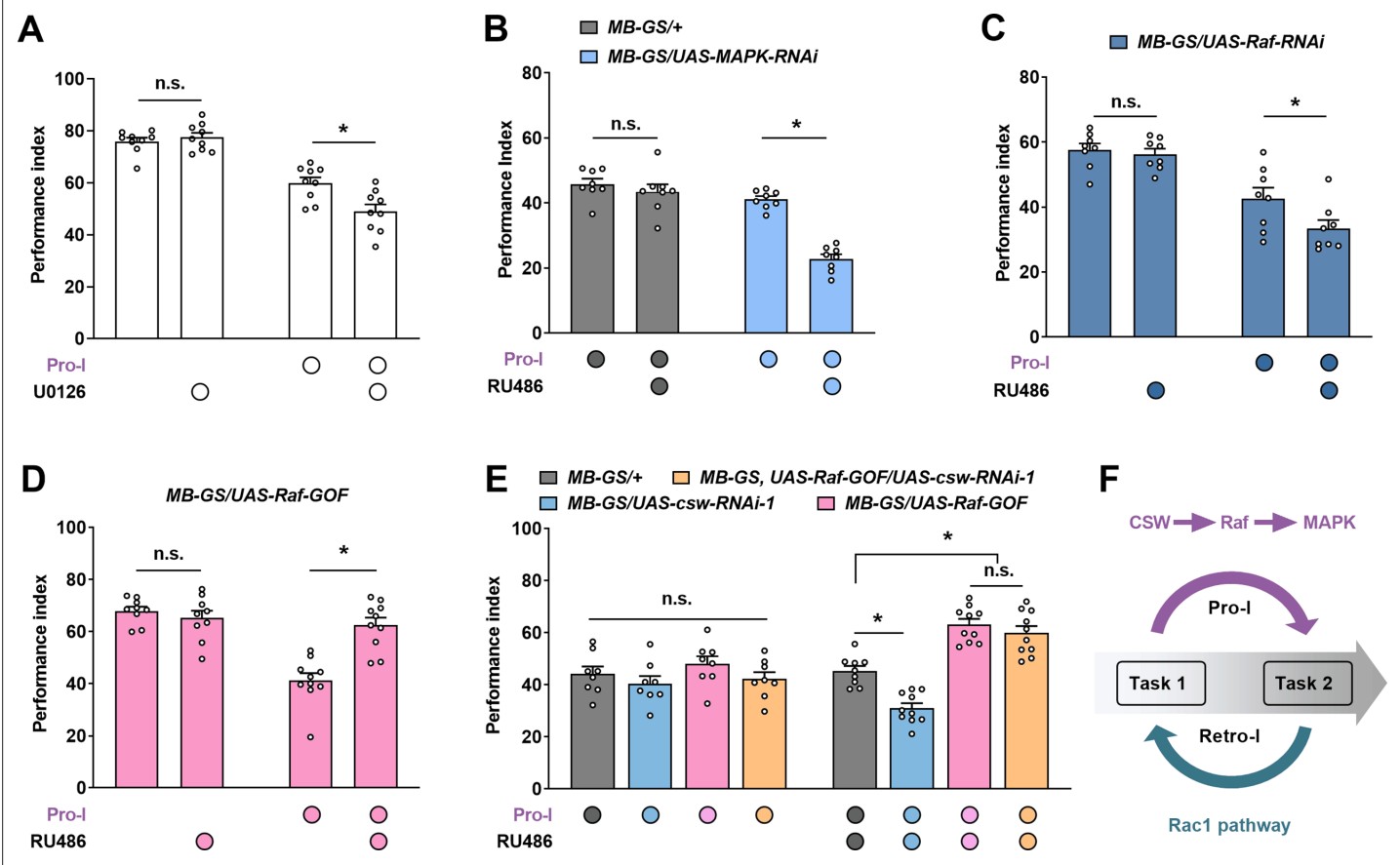

**Figure 4.** Corkscrew (CSW) regulates proactive interference (Pro-I) through Raf/MAPK pathway. The immediate memory performance with or without Pro-I (Pro-I, 0 min interval) was tested in wild-type and different transgenic flies (A–E). (A) Pharmacological inhibition of MAPK by feeding U0126 inhibitor aggravated the Pro-I in wild-type flies. n=9. (B) Flies with acute knockdown of MAPK in mushroom body (MB) neurons (MB-GS/UAS-MAPK-RNAi, RU486+) exhibited more severe Pro-I compared with uninduced control flies. n=8. (C) Acute knockdown of Raf in MB neurons (*MB-GS/UAS-Raf-RNAi*, RU486+) aggravated the Pro-I relative to the uninduced control. n=8. (D) Acutely overexpressing Raf-GOF in MB neurons (*MB-GS/UAS-Raf-GOF*, RU486+) reduced the Pro-I compared with its uninduced control group. n=9–10. (E) Acute overexpression of Raf-GOF (*MB-GS/UAS-Raf-GOF*, RU486+) dominated the effect of CSW knockdown (*MB-GS/UAS-csw-RNAi*, RU486+) on Pro-I. No significant difference was found between uninduced groups without RU486 feeding. n=8–10. (F) Model of molecular mechanisms underlying proactive and retroactive interference. Statistics: Mann-Whitney test (A and D); two-way ANOVA with Bonferroni's multiple comparisons tests (B, C, and E). Results with error bars are means ± SEM. *p<0.05. n.s., non-significant. Also see *Figure 4—figure supplement 1*, *Figure 4—source data 1*, and *Figure 4—figure supplement 1—source data 1* for additional information.

The online version of this article includes the following source data and figure supplement(s) for figure 4:

**Source data 1.** Raw data of *Figure 4*.

**Figure supplement 1.** Additional behavioral data of a single learning task and proactive interference (Pro-I).

**Figure supplement 1—source data 1.** Raw data of *Figure 4—figure supplement 1*.

---

1E). These results suggest that learning two different tasks consecutively may specifically recruit CSW to activate the Raf/MAPK pathway to reduce Pro-I through downstream molecules different from Sqh, a mechanism that differs from the memory protection mechanism triggered by a single learning task.

## Discussion

Our findings support a model for understanding molecular mechanisms of mutual interference between two successive learning tasks in *Drosophila* (*Figure 4F*). First, in learning two different associative tasks in succession, Pro-I and Retro-I occur simultaneously when the time distance between tasks is close (<20 min), while only Retro-I could be observed when the time distance was far (>20 min). Second, we identify a molecular pathway that specifically regulates Pro-I: the CSW/Raf/MAPK pathway. The

upstream signaling molecule CSW of this pathway is not involved in single-task learning and memory. In learning two different tasks close in time, although Pro-I and Retro-I occurred simultaneously, CSW was only involved in regulating Pro-I. Third, Rac1 specifically regulates Retro-I without affecting Pro-I when two different tasks are temporally close. Consistently, the Rac1/SCAR/Dia pathway has been reported to regulate Retro-I when the two tasks are temporally distant (1.5 hr interval) (*Gao et al., 2019*; *Shuai et al., 2010*).

According to our data, Pro-I is no longer evident when two aversive olfactory learning tasks are separated by more than 20 min (*Figure 2B and C*). After 20 min, the memory performance of the proactive task partially declines and a consolidated memory component called anesthesia-resistant memory (ARM) is largely formed (*Quinn and Dudai, 1976*; *Tully et al., 1994*; *Tully and Quinn, 1985*; *Zhang et al., 2016*). It raises a possibility that the decay or consolidation of the proactive task may release Pro-I. Based on this possibility, the reason why the CSW/Raf/MAPK pathway can bidirectionally regulate Pro-I can be explained because of the ability to modulate the decay or consolidation of proactive task memory. However, it is not supported by the following experimental results. First, acute knockdown or overexpression of CSW in MB neurons did not affect the immediate and early memory performance of a single task (*Figure 4—figure supplement 1B and C*). Second, in the Retro-I paradigm, the proactive task in the Pro-I paradigm became the target task and was directly examined, however, its performance was not affected by manipulating CSW (*Figure 3F*). Third, manipulating Raf/MAPK pathway in MB neurons during the adult stage did not affect immediate memory and ARM performance after a single aversive learning task (*Zhang et al., 2018*).

Our current findings suggest that inter-task similarity from different dimensions contributes to Pro-I. First, the content similarity between tasks can influence Pro-I. Two similar aversive associative tasks could produce significant Pro-I. However, Pro-I was not observed after reducing the inter-task content similarity by replacing the proactive task with non-associative stimuli or an appetitive associative task (*Figure 1A and B*). Due to the similarity of content, both tasks are more likely to use the same neural circuit thus increasing the possibility of Pro-I. Second, the similarity of environmental contexts between tasks plays an important role in Pro-I. By using different colors of light, we changed the environmental context similarity of the two tasks during the learning stage without affecting the content similarity. Pro-I was significantly released when the similarity of environmental contexts between tasks was reduced (*Figure 1C*). Third, the similarity of time contexts between tasks also contributes to Pro-I. Time is also considered as a context (*Bouton, 1993*). When the ITI was increased to more than 20 min, the Pro-I is no longer significant (*Figure 2A–C*). This phenomenon can be explained as the 20 min ITI might make the time contexts of the two tasks significantly different, thus reducing the inter-task similarity. According to this explanation, the CSW/Raf/MAPK pathway can regulate Pro-I probably by affecting time context similarity between tasks. Aversive learning can induce MAPK activation that peaks at around 20 min, which can be modulated by CSW and Raf (*Pagani et al., 2009*; *Zhang et al., 2018*). Overexpression of CSW in MB neurons is reported to accelerate MAPK activation (*Pagani et al., 2009*) and can reduce Pro-I (*Figure 3D, E and H*). Knocking down Raf in MB neurons shortens the duration of MAPK activation (*Zhang et al., 2018*) and can exacerbate Pro-I (*Figure 4C*). With these findings, an interesting idea is that MAPK activation can help distinguish the time contexts of the two tasks. Thus, the CSW/Raf/MAPK pathway can bidirectionally regulate Pro-I by affecting the similarity of time contexts between tasks. These properties of Pro-I that we obtained using the *Drosophila* model are consistent with psychological studies (*Kliegl and Bäuml, 2021*).

Although Pro-I in the current work only occurs when the interval between two tasks is less than 20 min, the phenomenon of Pro-I varies across biological learning systems and tasks (*Crossley et al., 2019*; *Epp et al., 2016*; *Jonides and Nee, 2006*). Future works are required to determine whether orthologues of CSW also participate in other forms of Pro-I. Interestingly, the molecules regulating Pro-I and Retro-I in *Drosophila* are involved in different diseases with varying levels of intellectual disability. Rac1 and Fmr1, which regulate Retro-I in *Drosophila,* are high-risk genes for autism (*Lord et al., 2020*). Mutations in *PTPN11*, a human orthologue of *csw*, account for more than half of Noonan syndrome (*Roberts et al., 2013*). Therefore, further studies on the molecular mechanisms underlying Pro-I and Retro-I may also contribute to the understanding of the pathogenesis of autism and Noonan syndrome. In addition, studies of Pro-I and Retro-I from the synaptic and neural circuit levels may also inspire continual learning in artificial intelligence (*Wang et al., 2021*; *Wang et al., 2022*).

## Materials and methods

### Fly stocks

Flies (*Drosophila melanogaster*) were cultured at 23°C and 60% relative humidity with standard medium under a 12 hr light-dark cycle. The standard medium consisted of 50 L of tap water, 1605 g of yeast, 530 g of agar, 438 g of potassium sodium tartrate, 36.3 g of calcium chloride, 1581 g of sucrose, 3160 g of glucose, 3885 g of corn flour, 142 g of preservatives, 1.875 g of penicillin, 1.875 g of chloramphenicol, 1.875 g of doxycycline, 3.71 g of amoxicillin, and 16.07 g of edible alkali. Flies using the TARGET system were raised at 18°C. *MB-GS* was a gift from Dr. Ronald L. Davis (*Mao et al., 2004*). *VT30604-Gal4* was a gift from Dr. Ann-Shyn Chiang (*Wu et al., 2013*). *Canton-S* (#64349), *UAS-Rac1-CA* (#6291), *UAS-Rac1-DN* (#6292), *UAS-csw* (#23878), *UAS-csw-RNAi-1* (#31760), *UAS-csw-RNAi-2* (#33619), *UAS-Raf-GOF* (#2033), *UAS-MAPK-RNAi* (#31524), *5-HT1B-Gal4* (#27637), *UAS-mCD8-::GFP* (#32186), and *Gal80^ts* (#7019 and 7017) were obtained from Bloomington Stock Center. The *UAS-Raf-RNAi* (#5796) and *UAS-sqh-RNAi* (#1223) were acquired from Tsinghua Fly Center. The *C739-Gal4* (*O'Dell et al., 1995*) was the extant stock in our lab.

### Generation of transgenic flies

Since the insertion site of *UAS-csw* in the transgenic line (#23878) acquired from Bloomington Storck Center is the X chromosome, a distinction between male and female flies needs to be made when counting behavioral results. For experimental convenience, we constructed a new transgenic strain (*UAS-csw*, insertion on chromosome III) and used it mainly in this study. Construction of this strain was performed at Fungene Biotech (http://www.fungene.tech). NotI/XbaI PCR fragment of coding sequences of *csw-RA* was inserted into the NotI/XbaI sites of pJFRC28-10XUAS-IVS-GFP-p10 vector (Addgene Plasmid #36431), and then the construct was inserted into *attp2* site.

### Behavioral assays

Flies from 2 to 6 days old were reared for behavioral experiments under the Pavlovian olfactory conditioning procedure (*Tully et al., 1994*; *Tully and Quinn, 1985*). Odors used were OCT (3-octanol, $1.5 \times 10^{-3}$ in dilution, Aldrich), MCH (4-methylcyclohexanol, $1.0 \times 10^{-3}$ in dilution, Fluka), EA (ethyl acetate, $2 \times 10^{-3}$ in dilution, Alfa Aesar), IA (isoamyl acetate, $2 \times 10^{-3}$ in dilution, Alfa Aesar), EL (Ethyl lactate, $1.5 \times 10^{-3}$ in dilution, Sigma-Aldrich), and PA (Pentyl acetate, $1.0 \times 10^{-3}$ in dilution, Sigma-Aldrich). OCT, MCH, EA, IA, EL, and PA were labeled as A, B, X, Y, E, and F in the figures, respectively. The flies were first transferred to a behavior room at 23 °C and 60% relative humidity for 30 min to adapt. The training time for each associative task was 5 min. About 100 flies were subjected to 90 s of air, 60 s of odor exposure accompanied by 12 pulses of electric shock at 60 V (conditioned stimulus +, CS+), 45 s of air, 60 s of another different odor (conditioned stimulus –, CS–) and 45 s of air. To test memory, trained flies were placed in a T-maze to choose between two odors, CS +and CS–. After 1 min of the choice, the performance index (PI) can be calculated based on the distribution of the flies between the two odors. A PI of 100 indicates that all flies escape CS +odor, while a PI of 0 means that flies have no preference for CS +and CS–. In particular, to test CS +or CS– memory components (*Figure 1—figure supplement 1C*), flies were allowed to choose between CS +and novel odor or between CS– and novel odor. For *Figure 1B*, an appetitive learning experiment was performed as described previously (*Yang et al., 2023*). Flies were placed in glass vials containing two pieces of 2 × 2 cm Watmann 3 MM filter paper soaked with distilled water and starved for 24 hr before training. The flies were then transferred to a behavior room at 23 °C and 60% relative humidity for 30 min to adapt. About 100 flies were sequentially exposed to a CS– odor for 2 min, air for 1 min, and a CS +odor accompanied by dry sucrose for 2 min. The memory was tested in the same way as aversive memory. For *Figure 1C*, the green or blue light context was provided by six manually controlled LEDs (12V5050RGB, Shenzhen Jinrui Photoelectric Co. LTD).

### Drug feeding treatment

Flies were fed with drugs as previously described (*Zhang et al., 2018*). Control flies (RU486– or U0126–) was fed a control solution containing 5% glucose and 3% ethanol. Flies were fed with 500 µM RU486 (Mifepristone, J&K) dissolved in a control solution in RU486 +groups. Flies were fed with 20 µM U0126 (Cell Signaling Technology) dissolved in a control solution for 16 hr before training.

## Transgene induction

In this study, two inducible systems, GeneSwitch (*Mao et al., 2004*) and TARGET (*McGuire et al., 2003*), were used for transgene expression. In the GeneSwitch system, the transgene expression was induced by RU486 feeding for two days. In the TARGET system, flies raised at 18°C were transferred to a 31°C incubator for three days to induce the transgene expression.

## Immunofluorescence

Adult flies were acutely ice anesthetized. Brains were dissected in ice-cold PBS (phosphate-buffered saline) and fixed in 4% paraformaldehyde for 55 min at room temperature. After three washes (10 min each) in PBT (0.5% Triton X-100 in PBS), samples were blocked in PBT containing 5% normal goat serum (NGS) for 90 min at room temperature. The brains were then transferred to a primary antibody solution (PBT containing the primary antibody and 5% NGS) and incubated for at least 24 hr at 4 °C. Chicken anti-GFP (1:1000, Abcam, Cat# ab13970; RRID: AB_300798) and mouse anti-Brp (1:10, nc82, DSHB; RRID: AB_2314866) were used as primary antibodies. Brains were then washed three times (10 min each) in PBT, transferred to secondary antibody solution (PBT with secondary antibody and 5% NGS), and incubated overnight at 4 °C. Goat anti-chicken IgG Alexa Fluor 488 (1:200, A-11039, Thermo Fisher Scientific; RRID:AB_2534096) and goat anti-mouse IgG Alexa Fluor 647 (1:200, A-21235, Thermo Fisher Scientific; RRID:AB_2535804) were used as secondary antibodies. Images were acquired using a Zeiss LSM710META confocal microscope and processed using Zen 2.6 blue edition.

## Statistics

Statistical analysis was performed using Prism (GraphPad). Normality tests were performed using D'Agostino and Pearson tests ($n \geq 8$) or Shapiro-Wilk tests ($n < 8$). For normally distributed data, comparisons between two groups were performed using the unpaired t-test, and comparisons of multiple groups were performed using ordinary one-way ANOVA with Dunnett's multiple comparisons test or two-way ANOVA with Bonferroni's multiple comparisons tests. For non-Gaussian distributed data, comparisons between two groups were performed using the Mann-Whitney test, and comparisons of multiple groups were performed using the Kruskal-Wallis test with Dunn's multiple comparison test. p-values <0.05 were considered statistically significant and marked with *, and n.s. means non-significant differences (p>0.05).

## Acknowledgements

We thank Dr. Ronald L Davis, Dr. Ann-Shyn Chiang, and the Bloomington Stock Center for fly stocks. We are grateful to Jun Zhou for his help in the experiments and Leo Shucheng Sun for his contribution in the discussion. This work was supported by grants from the National Natural Science Foundation of China (31970955, to QL; 32021002, to Yi Zhong) and the Tsinghua-Peking Center for Life Sciences.

## Additional information

### Funding

| Funder | Grant reference number | Author |
|---|---|---|
| National Natural Science Foundation of China | 31970955 | Qian Li |
| National Natural Science Foundation of China | 32021002 | Yi Zhong |
| Tsinghua-Peking Center for Life Sciences | | Yi Zhong |

The funders had no role in study design, data collection and interpretation, or the decision to submit the work for publication.

## Author contributions
Jianjian Zhao, Data curation, Formal analysis, Validation, Investigation, Visualization, Methodology, Writing - original draft, Writing - review and editing; Xuchen Zhang, Bohan Zhao, Conceptualization, Investigation, Methodology; Wantong Hu, Investigation, Visualization, Methodology, Writing - original draft, Writing - review and editing; Tongxin Diao, Investigation; Liyuan Wang, Writing - review and editing; Yi Zhong, Supervision, Funding acquisition; Qian Li, Conceptualization, Resources, Data curation, Formal analysis, Supervision, Funding acquisition, Visualization, Methodology, Writing - original draft, Project administration, Writing - review and editing

## Author ORCIDs
Bohan Zhao ⓘ http://orcid.org/0000-0002-9177-1278
Yi Zhong ⓘ http://orcid.org/0000-0002-7927-5976
Qian Li ⓘ http://orcid.org/0000-0001-7317-1570

## Decision letter and Author response
Decision letter https://doi.org/10.7554/eLife.83516.sa1
Author response https://doi.org/10.7554/eLife.83516.sa2

---

# Additional files

## Supplementary files
• MDAR checklist

## Data availability
All data generated or analysed during this study are included in the manuscript and supporting file.

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
