## [Editor Report]

This fundamental study substantially advances our understanding of interactions of consecutive memory tasks by identifying responsible molecules and neurons. The evidence supporting the claims of the authors is solid. The work will be of broad interest to neuroscientists working on learning and memory as well as learning psychologists.

---

## [Decision Letter]

**Decision letter after peer review:**

Thank you for submitting your article "Genetic Dissection of Mutual Interference between Two Consecutively Learned Tasks in *Drosophila*" for consideration by *eLife*. Your article has been reviewed by 2 peer reviewers, and the evaluation has been overseen by a Reviewing Editor and K VijayRaghavan as the Senior Editor. The following individuals involved in the review of your submission have agreed to reveal their identity: Emmanuel Perisse (Reviewer #1); Pedro F Jacob (Reviewer #2).

Essential revisions:

1) Appropriate controls and results description: To properly measure the effect of genetic manipulation, the performances of the experimental groups need to be compared to the corresponding parental control groups (cf. point 1 and last part of reviewers 1 and 2, respectively).

2) Putting the results of proactive interference into the context of existing literature: (related to point 2 and points 1-5 in reviewers 1 and 2, respectively). To address some of these points, further behavioural characterization is necessary.

*Reviewer #1 (Recommendations for the authors):*

1. In all the Figures, there is no UAS-… background control that must be used to conclude the effect of the genetic tool used. Why use only one background genetic control?

2. The authors could do some experiments and discuss a bit more on how (especially Pro) interference really happens. Considering the role of MB γ neurons in the acquisition of aversive memory their results are not surprising on the effect of proactive interference during memory retrieval between the trained odour versus the untrained odour. It will be interesting to understand a bit more about the proactive interference in the acquisition of both trained and untrained odours as they have meaningful information assigned during learning (negative and positive value, respectively; see Zhao et al., 2021; Nagano et al., 2022, and Jacob and Waddell, 2020). During the test, the authors could test each trained odour against a new odour. The reason why the performance index is low (or high) during the test may have different reasons. One of them could be that there is more (or less) generalization between the CS+ and the CS- in the trained task which they could test. Another could be that only CS+ and not CS- is learned which they could test. Why choose IA and EA as odours of interference learning? Do flies learn well this odour combination? Is the effect observed due to this odour combination? Can a different type of interference task induce the same proI effect, like appetitive learning?

*Reviewer #2 (Recommendations for the authors):*

In the current manuscript, the authors focus on a particular task of interference that might account for forgetting. As far as I am aware this is the first paper that tries to focus only on describing this task in more detail in *Drosophila*. However, this description of the phenomenon of Pro-I is limited and is focused only on applying previous knowledge. Therefore, in the current format, I believe that the manuscript requires more context (and experiments) within the interference literature.

Some examples:

1) In one of the senior author's (Yi Zhong) labs it was recently suggested that after a single conditioning event, two types of associative memories are encoded; one is context-independent (in the mushroom body) and the other is context-dependent (in the lateral horn). Substantial literature in the interference field has shown that Pro-I is a context-dependent phenomenon, and changing the context is a way to release from Pro-I. The authors in this manuscript show that the mushroom body is necessary for Pro-I, so how is this integrated with the fact that this memory is context-dependent but in the previous research of the authors, context is encoded in the lateral horn?

a. Further experiments are suggested to show that the observed task is indeed a classical example of Pro-I, and by changing the context of the two tasks, learning is no longer impaired. Furthermore, how is the manipulation of the expression of Corkscrew in γ Kenyon Cells influencing these tasks, after changing context?

2) The presence of Pro-I is circuit dependent both in invertebrates and vertebrates. Release of Pro-I can occur when two memories that require different circuits are sequentially learned since they can be consolidated in parallel. In *Drosophila* different types of memories (appetitive and aversive) require different circuits and can be formed in parallel.

a. Is Pro-I in *Drosophila*, also circuit-dependent? Given that aversive and appetitive memories require different compartments of the γ lobe of the mushroom body. This circuit dependency is a relevant part of the described work.

b. A more distinct proactive task from the target task is suggested to release Pro-I, since, on a posthoc basis, the target task can be more easily filtered out from the proactive set.

3) The observed Pro-I occurs within a time window (< 20 min), where the new learning (target task) occurs during the consolidation of the old memory (proactive task).

a. Contextualization within the memory consolidation is required. How passive decay of the memory for the proactive task might influence the reduced impairment of the target task in longer time intervals.

4) Over decades an important debate is how to explain Pro-I effect, some attribute Pro-I to a problem at the retrieval stage and others to a problem at the encoding stage.

a. Further contextualization of the current findings within this debate would be interesting.

5) A more general question, which stimulus/task dimensions (e.g. lag between tasks; different lengths of memory, short, mid, long-term; similarity of stimulus/tasks) influence Pro-I in *Drosophila*?

The manuscript would benefit from a brief introduction for the general audience of the tasks used at the beginning of the Results section.

In line 122. It should read Figure 2G instead of Figure 3G.

The overall reporting of the statistics needs to be improved across the manuscript, for example in lines 148 to 150 it is stated:

"Flies with acute genetic knockdown of MAPK in MB neurons (MB-GS/UAS-MAPK-RNAi, RU486+) also exhibited more severe Pro-I than uninduced and parental control flies (Figure 4B)."

The description of the statistics is always mentioned in this way in the manuscript, which might lead to erroneous interpretations from the readers. For example, here the MB-GS/UAS-MAPK-RNAi induced with RU486 after Pro-I was compared to MB-GS/UAS-MAPK-RNAi induced with RU486 without Pro-I, and never to the parental controls.

In figure 3, the authors start to present the data of overexpressing Corkscrew constitutively in the different subpopulations of Kenyon cells (Figure 3a) and correctly try to rule out possible development effects by only overexpressing Corkscrew in the adult. The authors conclude that the initial effect in γ Kenyon Cells is adult-specific (Figure 3b), whereas the effect in the α/β Kenyon cells is not (Figure 3c). All the manipulations should have been done in the adult only (additionally α′/β′ Kenyon Cells possibly, for sake of completion), and the presentation of the data in Figure 3a is no longer relevant because the initial effect in the α/β Kenyon cells is not reproduced in the adult. Therefore, Figure 3a should be removed, and completed with adult-specific manipulation of α′/β′ Kenyon Cells.

Given these effects during development after overexpression of Corkscrew, appropriate sensory acuity tests (for shock and odour) should be presented. This should be done for the genotypes that show statistical differences.

The n of 6/7 in some of the experiments is below the typical in the field (n=8 to 12).

The formatting of some references is different from the majority. For example Bouton (1993); Brand and Perrimon (1994), Luo et al., (1994), and McGuire et al., (2003).

---

## [Author Response]

Reviewer #1 (Recommendations for the authors):1. In all the Figures, there is no UAS-… background control that must be used to conclude the effect of the genetic tool used. Why use only one background genetic control?

For inducible systems of transgene expression (e.g. GeneSwitch), we thought that uninduced flies (e.g. RU486-) are better controls relative to genetic controls (e.g. MB-GS/+ or UAS controls) because their genetic background is not altered and the main alteration is attributed to the induced expression of the transgene (RU486 feeding). Therefore, in the initial manuscript, we used only one type of genetic control group (*MB-GS/+*) when investigating the role of CSW in proactive interference.

To further strengthen our data about CSW, we performed four new experiments in revision. For experiments using the GeneSwitch system, as the reviewer suggested, we added UAS control groups (*+/UAS-csw* and *+/UAS-csw-RNAi-1*) and performed new experiments (revised Figure 3—figure supplement 1D and 1E). For experiments using the TARGET system, we added new data including a CSW-knockdown experiment (revised Figure 3I) and an uninduced experiment (revised Figure 3—figure supplement 1H). All these new results further strengthened our findings in the initial manuscript.

For the experimental data related to Rac1, MAPK, and Raf, we used the same fly strains as in our previous studies (Shuai, et al., Cell, 2010; Zhang, et al., Neuron, 2018; Gao, et al., PNAS, 2019; Mo, et al., Aging Cell, 2022) and did not add additional UAS control experiments.

2. The authors could do some experiments and discuss a bit more on how (especially Pro) interference really happens. Considering the role of MB γ neurons in the acquisition of aversive memory their results are not surprising on the effect of proactive interference during memory retrieval between the trained odour versus the untrained odour. It will be interesting to understand a bit more about the proactive interference in the acquisition of both trained and untrained odours as they have meaningful information assigned during learning (negative and positive value, respectively; see Zhao et al., 2021; Nagano et al., 2022, and Jacob and Waddell, 2020). During the test, the authors could test each trained odour against a new odour. The reason why the performance index is low (or high) during the test may have different reasons. One of them could be that there is more (or less) generalization between the CS+ and the CS- in the trained task which they could test. Another could be that only CS+ and not CS- is learned which they could test. Why choose IA and EA as odours of interference learning? Do flies learn well this odour combination? Is the effect observed due to this odour combination? Can a different type of interference task induce the same proI effect, like appetitive learning?

First, as suggested by the reviewer, we tested each trained odor against a new odor to obtain CS+ and CS- memory performance, respectively (Revised Figure 1—figure supplement 1C). Consistent with previous reports (Jacob and Waddell, Neuron, 2020; Zhao et al., *eLife*, 2021; Nagano et al., Eur J Neurosci, 2022), CS+ memory was avoidance memory, while CS- memory was approach memory. Interestingly, Pro-I significantly affected CS+ memory performance, but not CS- memory performance. This result further enriches our understanding of the Pro-I. Second, since our previous studies (Shuai et al., Cell, 2010; Dong et al., PNAS, 2016; Zhang et al., Cell Rep, 2016; Gao et al., PNAS, 2019) have used EA and IA as odors for the interference task, they were not intentionally changed in our current work. According to our newly added experimental data (Revised Figure 1—figure supplement 1B), the use of EA and IA odors in the target task also produced high learning performance and can be significantly affected by the proactive task using odors of OCT and MCH. Therefore the Pro-I effect was not due to the use of a specific odor combination. Third, we performed the experiment suggested by the reviewer and found that there was no significant Pro-I when appetitive learning was used as a proactive task and aversive learning was used as a target task (Revised Figure 1B).

Reviewer #2 (Recommendations for the authors):In the current manuscript, the authors focus on a particular task of interference that might account for forgetting. As far as I am aware this is the first paper that tries to focus only on describing this task in more detail in *Drosophila*. However, this description of the phenomenon of Pro-I is limited and is focused only on applying previous knowledge. Therefore, in the current format, I believe that the manuscript requires more context (and experiments) within the interference literature.Some examples:1) In one of the senior author's (Yi Zhong) labs it was recently suggested that after a single conditioning event, two types of associative memories are encoded; one is context-independent (in the mushroom body) and the other is context-dependent (in the lateral horn). Substantial literature in the interference field has shown that Pro-I is a context-dependent phenomenon, and changing the context is a way to release from Pro-I. The authors in this manuscript show that the mushroom body is necessary for Pro-I, so how is this integrated with the fact that this memory is context-dependent but in the previous research of the authors, context is encoded in the lateral horn?a. Further experiments are suggested to show that the observed task is indeed a classical example of Pro-I, and by changing the context of the two tasks, learning is no longer impaired. Furthermore, how is the manipulation of the expression of Corkscrew in γ Kenyon Cells influencing these tasks, after changing context?

In our regular experiments to study aversive olfactory memory, the learning context of flies (tubes with copper grid) is different from the test context (tubes without copper grid). Under such conditions, the lateral horn-related, context-dependent memory discovered in the previous study (Zhao, et al., Nature Communications, 2019) cannot be detected. This context-dependent memory can be observed only when the test context is consistent with the learning context (e.g., by adding the copper grid to the test tubes to make the test and learning context consistent). Since all experiments in our current manuscript were performed using the regular method (learning and test context are different), the context-dependent memory in the lateral horn was not involved in our findings.

The suggestion to change the context of the two tasks is very constructive. Since flies are trained in a dark room in regular behavioral experiments, the proactive task and the target task in Pro-I experiments had the same dark context. We can still observe the same Pro-I phenomenon after changing the context of both tasks from dark to blue light. However, when the contexts of the two tasks become different (green light context for the proactive task and blue light context for the target task), the Pro-I could no longer be observed (revised Figure 1C). In psychological studies, Pro-I can be released by making learning contexts of target and non-target tasks more distinct (Kliegl and Bauml, Neuroscience and Biobehavioral Reviews, 2021). Our new result is in line with these findings, suggesting that the observed result is indeed a classical example of Pro-I. Interestingly, when the contexts of the two tasks become different, knocking down CSW in MB neurons did not increase the Pro-I (revised Figure 3—figure supplement 1I). In contrast, the same manipulation significantly aggravated the Pro-I when it was released by increasing the ITI to 20 min (revised Figure 3C).

2) The presence of Pro-I is circuit dependent both in invertebrates and vertebrates. Release of Pro-I can occur when two memories that require different circuits are sequentially learned since they can be consolidated in parallel. In *Drosophila* different types of memories (appetitive and aversive) require different circuits and can be formed in parallel.a. Is Pro-I in *Drosophila*, also circuit-dependent? Given that aversive and appetitive memories require different compartments of the γ lobe of the mushroom body. This circuit dependency is a relevant part of the described work.b. A more distinct proactive task from the target task is suggested to release Pro-I, since, on a posthoc basis, the target task can be more easily filtered out from the proactive set.

As suggested by the reviewer, we used an appetitive learning task as a significantly different proactive task from the target task (revised Figure 1B). No significant proactive interference was observed. In other words, proactive interference was released, supporting that Pro-I in *Drosophila* is also circuit-dependent.

3) The observed Pro-I occurs within a time window (< 20 min), where the new learning (target task) occurs during the consolidation of the old memory (proactive task).a. Contextualization within the memory consolidation is required. How passive decay of the memory for the proactive task might influence the reduced impairment of the target task in longer time intervals.

According to our data, Pro-I is no longer evident when two aversive olfactory learning tasks are separated by more than 20 min (revised Figures 2B and 2C). After 20 min, the memory performance of the proactive task partially declines and a consolidated memory component called anesthesia-resistant memory (ARM) is largely formed (Quinn and Dudai, Nature, 1976; Tully and Quinn, Journal of Comparative Physiology a-Sensory Neural and Behavioral Physiology, 1985; Tully, et al., Cell, 1994; Zhang, et al., Cell Reports, 2016). It raises a possibility that the decay or consolidation of the proactive task may release Pro-I. Based on this possibility, the reason why the CSW/Raf/MAPK pathway can bidirectionally regulate Pro-I can be explained because of the ability to modulate the decay or consolidation of proactive task memory. However, it is not supported by the following experimental results. First, acute knockdown or overexpression of CSW in MB neurons did not affect the immediate and early memory performance of a single task (revised Figure 4—figure supplement 1B and 1C). Second, in the Retro-I paradigm, the proactive task in the Pro-I paradigm became the target task and was directly examined, however, its performance was not affected by manipulating CSW (revised Figure 3F). Third, manipulating Raf/MAPK pathway in MB neurons during the adult stage did not affect immediate memory and ARM performance after a single aversive learning task (Zhang, et al., Neuron, 2018). We have added a related discussion in the revised manuscript.

4) Over decades an important debate is how to explain Pro-I effect, some attribute Pro-I to a problem at the retrieval stage and others to a problem at the encoding stage.a. Further contextualization of the current findings within this debate would be interesting.

Regarding whether Pro-I is attributed to a problem at the retrieval stage or a problem at the encoding stage, we currently do not have a preferential answer based on the available data. Our means of manipulating the CSW/MAPK pathway could theoretically affect both the encoding and retrieval stages. In the future, if there is a very rapid tool to manipulate CSW/MAPK pathway, more clear evidence can be obtained. Alternatively, studies of Pro-I at the neural circuit level may also yield useful evidence.

5) A more general question, which stimulus/task dimensions (e.g. lag between tasks; different lengths of memory, short, mid, long-term; similarity of stimulus/tasks) influence Pro-I in *Drosophila*?The manuscript would benefit from a brief introduction for the general audience of the tasks used at the beginning of the Results section.

As suggested, we added a brief introduction at the beginning of the Results section in the revised manuscript.

In line 122. It should read Figure 2G instead of Figure 3G.

We have corrected this error.

The overall reporting of the statistics needs to be improved across the manuscript, for example in lines 148 to 150 it is stated:"Flies with acute genetic knockdown of MAPK in MB neurons (MB-GS/UAS-MAPK-RNAi, RU486+) also exhibited more severe Pro-I than uninduced and parental control flies (Figure 4B)."The description of the statistics is always mentioned in this way in the manuscript, which might lead to erroneous interpretations from the readers. For example, here the MB-GS/UAS-MAPK-RNAi induced with RU486 after Pro-I was compared to MB-GS/UAS-MAPK-RNAi induced with RU486 without Pro-I, and never to the parental controls.

We apologize for these misleading descriptions. We described them this way because comparisons with the parental control (MB-GS/+) were also done in the analysis of the experimental data, but were not shown in the figures. For the data in Figure 4B, we made the figure in a different way (revised Figure 4—figure supplement 1A). Acutely knocking down MAPK (*MB-GS/UAS-MAPK-RNAi*, RU486+) led to more severe Pro-I compared to the genetic control (MB-GS/+, RU486+).

In addition, we have made the following two changes in order to present our data more clearly. First, we presented the data of Figure 3B, Figure 3C and Figure 3E in a different way (revised Figure 3—figure supplement 1A-1C). Second, we performed new experiments with another genetic control group (revised Figure 3—figure supplement 1D and 1E). All the added results were consistent with our findings.

In figure 3, the authors start to present the data of overexpressing Corkscrew constitutively in the different subpopulations of Kenyon cells (Figure 3a) and correctly try to rule out possible development effects by only overexpressing Corkscrew in the adult. The authors conclude that the initial effect in γ Kenyon Cells is adult-specific (Figure 3b), whereas the effect in the α/β Kenyon cells is not (Figure 3c). All the manipulations should have been done in the adult only (additionally α′/β′ Kenyon Cells possibly, for sake of completion), and the presentation of the data in Figure 3a is no longer relevant because the initial effect in the α/β Kenyon cells is not reproduced in the adult. Therefore, Figure 3a should be removed, and completed with adult-specific manipulation of α′/β′ Kenyon Cells.Given these effects during development after overexpression of Corkscrew, appropriate sensory acuity tests (for shock and odour) should be presented. This should be done for the genotypes that show statistical differences.

A previous study (Pagani, et al., Cell, 2009) has shown that overexpression of CSW in all KCs or α/β KCs throughout developmental and adult stages does not affect electric shock reactivity and odor acuity. So we did not do similar experiments again. We first examined the Pro-I effect by constitutively overexpressing CSW in different KC subpopulations (Figure 3A) mainly because of the following three reasons. First, we did not have GeneSwitch tools specific to KC subpopulations including γ, α/β, and α’/β’ KCs. Second, for TARGET tools, we only have available combined fly strains for γ (Gal80^ts^; 5-HT1B) and α/β KCs (C739; Gal80^ts^) but not α’/β’ KCs. Third, according to the previous study (Pagani, et al., Cell, 2009), manipulating CSW in KCs with or without bypassing the developmental stage had similar effects in regulating long-term memory formation. Considering this review suggestion together with Reviewer #1’s suggestion, we moved Figure 3A to supplemental data (revised Figure 3—figure supplement 1G). At the same time, we performed new experiments to further strengthen our previous experimental data using the TARGET system. We added new data including a CSW-knockdown experiment (revised Figure 3I) and an uninduced experiment (revised Figure 3—figure supplement 1H).

The n of 6/7 in some of the experiments is below the typical in the field (n=8 to 12).

In revision, we increased the N of Figure 1F (revised Figure 2F, from 8 to 12), (Figure 2D revised Figure 3C, from 6 to 10), and Figure 2G (revised Figure 3F, from 6 to 10), respectively. The N for all revised figures is 8 or more, except for Figure 3E (n = 7-8), Figure 3—figure supplement 1H (n = 7-8), Figure 4—figure supplement 1B (n = 7) and 1D (n = 7).

The formatting of some references is different from the majority. For example Bouton (1993); Brand and Perrimon (1994), Luo et al., (1994), and McGuire et al., (2003).

We have made revisions to make the references in the same format.